# Self-Supervised Cortical Surface Reconstruction for Ultra High-resolution *ex vivo* 7T MRI

**Haoxiang Li**[*1]                                LIHAOXIA24@MAILS.TSINGHUA.EDU.CN
**Mingxuan Liu**[*1]                               ARKTISX@FOXMAIL.COM
**Hongjia Yang**[1]                                YANGHJ23@MAILS.TSINGHUA.EDU.CN
**Yi Liao**[2]                                     CONNIE0064@126.COM
**Haibo Qu**[2]                                    WINDOWSQHB@126.COM
**Jonathan Polimeni**[3]                           JRPOLIMENI@STANFORD.EDU
**Qiyuan Tian**[†1]                                QIYUANTIAN@TSINGHUA.EDU.CN

[1] *School of Biomedical Engineering, Tsinghua University*

[2] *Department of Radiology, West China Second University Hospital, Sichuan University*

[2] *Department of Radiology, Stanford University*

**Editors:** Accepted for publication at MIDL 2025

## Abstract

*Ex vivo* brain MRI enables sub-millimeter ultra-high-resolution studies, uncovering structural details unattainable with *in vivo* MRI. Cortical surface reconstruction (CSR) based on these detailed images is crucial for studying cortical anatomy and structure. In this study, we propose **SelfCSR**, a self-supervised deep learning framework for accurate *ex vivo* 7T MRI CSR without the need for manually labeled training data.

**Keywords:** *Ex vivo* Brain, 7T MRI, Self-supervised Learning, Cortical Surface Reconstruction.

## 1. Introduction

*Ex vivo* MRI offers significant advantages over *in vivo* MRI by enabling detailed neuroanatomy visualization, bridging microscale histology studies with morphometric measurements, and linking macroscopic features such as cortical thickness to underlying cytoarchitecture and pathology (Khandelwal et al., 2024). Meanwhile, compared to standard 1.5T or 3T MRI, 7T MRI provides ultra-high resolution and significantly enhanced contrast, making it an invaluable tool for detailed neuroimaging. Since *ex vivo* MRI can be conducted with less time constraints (e.g., lasting days) and is free from cardiorespiratory or head motion, it enables the application of 7T imaging for brain neuroscientific studies. For example, Coras et al. (Coras et al., 2014) leveraged 7T *ex vivo* MRI to characterize the microstructural differences between normal and sclerotic hippocampi in temporal lobe epilepsy. Zeng et al. (Zeng et al., 2024) developed a segmentation model specifically for 7T *ex vivo* MRI data using manually labeled training datasets. Another study also developed a deep learning pipeline for automated high-resolution postmortem MRI segmentation, linking cortical and subcortical morphometry to neuropathology in neurodegenerative diseases (Khandelwal et al., 2024).

---

[*] Contributed equally

[†] Corresponding author

Cortical surface reconstruction (CSR) for *ex vivo* brain is crucial for studying cortical anatomy and structure. However, existing CSR tools, such as FreeSurfer (Fischl, 2012) and BrainSuite (Shattuck and Leahy, 2002), struggle to process 7T MRI *ex vivo* brain data due to its unique tissue contrast, ultra-high resolution, and geometric distortions arising from the absence of CSF and menings. These violations of in vivo assumptions, such as opposing sulcal banks being pressed together, complicate tissue segmentation and cortical surface reconstruction.

Deep learning provides a promising approach for CSR (Ma et al., 2022; Li et al., 2025). However, due to the scarcity of datasets, limited imaging resources, pronounced susceptibility artifacts, and signal inhomogeneity, existing supervised learning methods like Cortex-ODE (Ma et al., 2022) and CoTAN (Ma et al., 2023) are not applicable to *ex vivo* brain imaging. Recent advances, such as SegCSR (Zheng et al., 2024) and CoSeg (Ma et al., 2024), have explored self-supervised CSR for *in vivo* T1-weighted brain MRI by leveraging pseudo ground truth synthesized from segmentation. Despite these advancements, there remains a lack of methods tailored for cortical surface reconstruction directly from 7T MRI data. To address this gap, we present **SelfCSR**, the first self-supervised CSR method tailored for 7T *ex vivo* brain MRI. **SelfCSR** harnesses the unique advantages of 7T MRI while addressing challenges from its distinct tissue contrast and high resolution, enabling more accurate and efficient cortical surface analysis.

## 2. Method

### 2.1. Data Description

This study used the publicly available 7 Tesla *ex vivo* human brain MRI dataset at 100-micron resolution from Edlow et al. (Edlow et al., 2019). The scans were acquired using a 7 Tesla whole-body human MRI scanner with four single-echo spoiled gradient-recalled echo (SPGR/GRE) or Fast Low-Angle Shot (FLASH) sequences. The dataset features exceptionally high-resolution images (1600×1400×640 voxels), with each dataset approximating 4.9GB in size.

### 2.2. *ex vivo* Brain Cortex Surface Reconstruction Pipeline

The proposed **SelfCSR** consists of four stages: automatic segmentation, downsampling and denoising, marching cubes, and self-supervised surface deformation.

**Automatic Segmentation**. The segmentation of supragranular and infragranular layers is achieved using the semi-supervised multi-resolution U-Net framework proposed by Zeng et al (Zeng et al., 2024).

**Downsampling and Denoising**. The data is first downsampled to an isotropic resolution of 0.5 mm, significantly reducing memory consumption while preserving essential anatomical details. Furthermore, the BM4D algorithm (Maggioni et al., 2012) is applied for denoising to enhance data quality.

**Marching Cube**. The marching cubes algorithm (WE, 1987) is employed to extract three cortical surfaces from the segmentation: the white matter (WM) surface, the mid-surface (representing the boundary between the supragranular and infragranular layers), and the pial surface. However, the middle surface and the pial surface are different from

the true boundary due to partial volume effects and possible topological errors. The WM surface is used as the initial input and reference geometry for subsequent deformation, while the mid-surface and pial surface are used as pseudo-labels for training in the framework.

**Surface Deformation**. To ensure point-wise correspondence between the three generated surfaces, we trained two U-Net models to predict stationary velocity fields (SVFs). The first U-Net deforms the white matter surface to align with the mid-surface, and the second U-Net subsequently deforms the mid-surface to align with the pial surface. Inspired by FreeSurfer's intensity term and spring term(Dale et al., 1999), we designed a self-supervised loss function:

$$\mathcal{L} = w_{cd}\mathcal{L}_{cd} + w_{gr}\mathcal{L}_{gr} + w_{it}\mathcal{L}_{it} + w_{edge}\mathcal{L}_{edge} + w_{normal}\mathcal{L}_{normal} \tag{1}$$

The boundary alignment loss ($\mathcal{L}_{cd}$) optimizes surface-to-boundary fit using one-way Chamfer distance(Zheng et al., 2025), while the gradient loss ($\mathcal{L}_{gr}$) maximizes MRI gradients at the surface for sharper boundaries. The intensity loss ($\mathcal{L}_{it}$) constrains voxel values to plausible ranges. Topological integrity is enforced by a mesh regularity term ($\mathcal{L}_{edge}$) penalizing irregular edge lengths and an inflation constraint ($\mathcal{L}_{deform}$) ensuring normal-direction expansion. The terms $w_{cd}$, $w_{gr}$, $w_{it}$, $w_{edge}$, and $w_{normal}$ serve as weight hyperparameters to balance the contribution of each loss component.

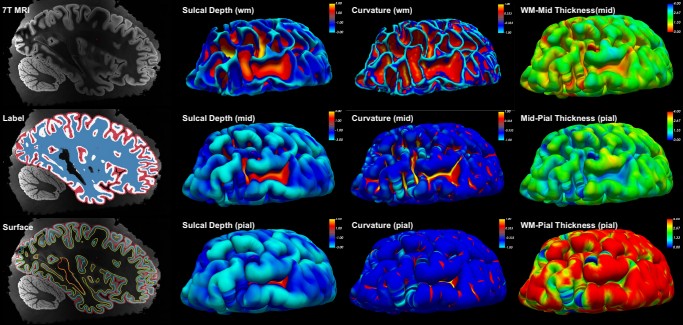

Figure 1: Cortical surface reconstruction and morphological analysis. Partial defects exist on the pial surface.

## 3. Results and Conclusion

Figure 1 demonstrates the reconstruction results of the corresponding WM surface, mid surface, and pial surface. The model accurately identifies the boundaries between different cortical layers. We further analyzed *ex vivo* brain morphology using FreeSurfer (Fischl, 2012) to quantify cortical curvature, thickness, and sulcal depth. In conclusion, **SelfCSR** is the first learning-based *ex vivo* CSR method and offering a novel tool for *ex vivo* MRI-based neuroimaging research.

## Acknowledgments

Tsinghua University Startup Fund and Dushi Program (grant number 20241080026).

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
