# OpenReview forum: "Self-Supervised Cortical Surface Reconstruction for Ultra High-resolution ex vivo 7T MRI"
_MIDL.io/2025/Short_Papers — MIDL 2025 - Short Papers_

### Official Review · Reviewer_oC62 · 2025-04-20

**Rating:** 4
**Confidence:** 4

**Summary:**

This paper introduces a framework for cortical surface reconstruction from 7T MRI. The process involves four steps: segmentation, denoising, marching cubes, and surface registration. Visualizations show that the resulting surfaces are satisfactory.

**Strengths:**

1. The authors should be commended for developing a tool for a novel application.
2. The paper is self-contained.

**Weaknesses:**

1. For a MIDL paper, I expected more technical novelty. While the authors claim their tool is novel, it is unclear which part specifically addresses the mentioned challenges (e.g., limited resources). Each step appears to have been previously discussed in the literature. The paper lacks effort in connecting the motivation and methods.
2. The term "self-supervised learning" is misleading—the procedure seems conventional, following a fully supervised learning scheme with regularization.

---

### Decision · Program_Chairs · 2025-05-01

Accept